

# Agricultural water allocation with climate change based on gray wolf optimization in a semi-arid region of China

Zhidong Wang[1], Xining Zhao[1,2], Jinglei Wang[3], Ni Song[3] and Qisheng Han[3]

[1] College of Water Resources and Architectural Engineering, Northwest A & F University, Yangling, China
[2] Institute of Soil and Water Conservation, Northwest A&F University, Yangling, China
[3] Farmland Irrigation Research Institute of Chinese Academy of Agriculture Sciences/Key Laboratory of Crop Water Use and Regulation, Ministry of Agriculture and Rural affairs, Xinxiang, China

## ABSTRACT

**Background**. We quantified and evaluated the allocation of soil and water resources in the Aksu River Basin to measure the consequences of climate change on an agricultural irrigation system.

**Methods**. We first simulated future climate scenarios in the Aksu River Basin by using a statistical downscaling model (SDSM). We then formulated the optimal allocation scheme of agricultural water as a multiobjective optimization problem and obtained the Pareto optimal solution using the multi-objective grey wolf optimizer (MOGWO). Finally, optimal allocations of water and land resources in the basin at different times were obtained using an analytic hierarchy process (AHP).

**Results**. (1) The SDSM is able to simulate future climate change scenarios in the Aksu River Basin. Evapotranspiration ($ET_0$) will increase significantly with variation as will the amount of available water albeit slightly. (2) To alleviate water pressure, the area of cropland should be reduced by 127.5 km$^2$ under RCP4.5 and 377.2 km$^2$ under RCP8.5 scenarios. (3) To be sustainable, the allocation ratio of forest land and water body should increase to 39% of the total water resource in the Aksu River Basin by 2050.

## INTRODUCTION

Economic and social development are constrained by various factors, including shortages of available water and land resources, climate change and environmental degradation (*Bai et al., 2015*). The IPCC's Sixth Assessment Report (AR6) has made clear that climate change is intensifying the water cycle and affecting rainfall patterns (*IPCC, 2021*), which will have a significant impact on the global hydrological cycle and water balance (*Miller & Belton, 2014*). With rapid population and economic growth, it is difficult to recocile trade-offs among water and land management, ecological environmental protection, and socio-economic development (*Mei et al., 2010*).

As the world population and consequent demand for food increase, safe water for agricultural use has become increasingly scarce (*Summerlin et al., 2021*). This phenomenon

Corresponding authors
Jinglei Wang, wangjinglei@caas.cn
Qisheng Han, hanqisheng@caas.cn

is pronounced in arid and semiarid regions with irrigation (*Rasouli, Kiani Pouya & Cheraghi, 2012*). The total amount of argicultural irrigation is often determind by planting area and planting structure. However, ET is usually calculated as a crop coefficient in irrigation planning with evapotranspiration ($ET_0$) as a key indicator (*Wu et al., 2021*).

Recent studies have found that $ET_0$ should change with the effects of climate change, especially where agricultural water consumption accounts for a large proportion of use (*Zou et al., 2020*). Our ability to accurately simulate future climatic scenaros will be the basis for estimating the $ET_0$. Although future climate conditions can be roughly estimated using general circulation models (GCMs), meeting requirements for high resolution has been challenging (*Wilby, Dawson & Barrow, 2002*). Therefore, it is necessary to use downscaling methods to ''shrink'' the study area to specific areas or sites for practical application (*Hewitson & Crane, 2006*).

There are two dominant downscaling approaches: dynamic and statistical. Statistical downscaling is widely used because of its simple operation and low cost (*Vallam & Qin, 2018*). A statistical downscaling model (SDSM) is used to produce the required high-resolution climate projection by developing a statistical relationship between the large- and local-scale climate variables (*Gebrechorkos, Hülsmann & Bernhofer, 2019*). As such it is more stuitable for climate change simulation at local scales. In previous studies, the impact of climate change on regional inflow and demand and the feedback relationship between supply and demand were ignored (*Fu et al., 2014*). For example, *Sun et al. (2018)* have considered the impact of climate change on watershed runoff, but ignored the impact of different climatic conditions on agricultural and ecological water demand. Therefore, how to simultaneously consider changes in water resources and water demand under climate change and realize a balanced allocation of regional water and land resources is a problem that needs to be solved urgently.

However, industrial/domestic and ecological water are considered equally important for regional development. Water and land optimization allocation is also a complex problem that involves many elements (*Habibi Davijani et al., 2016*). How to rationalize planting structure with the effects of climate change is an important consideration in water resource management. The best way to solve this problem is to build a multi-objective model for optimization. Approaches such as evolutionary (EA), genetic (GA), and nondominated sorting genetic algorithms (NSGA-II), linear (LP) and non-linear programming (NLP), among others, have been applied to optimize water and land resources (*Keshtkar et al., 2020*). These methods provide multiple options for decision makers by finding a model Pareto solution set. However, most have a number of shortcomings, (*e.g.*, local optima traps or slow convergence). The grey wolf optimizer (GWO) algorithm, which was proposed by *Mirjalili, Mirjalili & Lewis (2014)*, is a relatively novel population-based metaheuristic algorithm that combines fast convergence and high optimization accuracy (*Rashidi et al., 2018*). The GWO algorithm utilizes the simulated social leadership and encircling mechanism in order to find the optimal solution for single-objective optimization problems (*Mirjalili, Mirjalili & Lewis, 2014*). For preforming multi-objective optimization, the multi-objective GWO (MOGWO) extends the advantages of GWO to more complex scenarios. However, the shortcomings of GWO (initial value effects, local optimum traps)

when solving multi-objective problems have been improved (*Mirjalili et al., 2016*). Then, the analytic hierarchy process (AHP) is used to select the most suitable options from the Pareto solutions. AHP was an analytic technique for multiobjective decesions combined with qualitative and quantitative analysis, and determines the weights of factors by using the multifactor classification method (*Di et al., 2018*). Water demand allocation would be qualitative rather than quantitative, which was an advantage of the AHP method.

Xinjiang Province is a typical of arid and semi-arid region in China that lacks significant water resources, and the surface runoff are primarily generated by glacier meltwater in adjacent mountains (*Chen et al., 2020*). Water shortages have become a source of conflict in the Tarim River Basin of Southern Xinjiang with intense confrontations between environmental protection and economic development (*Lam, Kleinn & Coenradie, 2011*). As an ecologically fragile area, the Tarim River Basin has experienced a significant decline of its riparian desert forests (*Zhang et al., 2019*). Additionally, the Tarim River Basin is a major source of cotton and fruit production. Therefore, it is particularly important to improve water use efficiency and optimize allocation of water resources in this region.

This study improves on previous research as follows: (1) we considered the case that crop water requirements were not fixed but rather varies with climate change; (2) the machine learning method was used to estimate the runoff with climate data in the future; (3) this study was a novel attempt to solve the problem of water conflicts by integrating the AHP and multi-objective GWO. The main aims of our study were the following aspects: (1) forecast regional climate change scenarios using the SDSM model; (2) calculate regional water supply and demand in different climate scenarios; (3) determine water consumption among crops and establish a multi-objective programming model using the MOGWO algorithm to solve water-use conflicts for agricultural production, ecosystems, and drinking water supply; (4) select the most suitable options from the Pareto solutions using an analytic hierarchy process (AHP).

## MATERIALS & METHODS

### Study area

The study area was the Aksu Valley (75°35′–82°00′E, 40°00′–42°27′N, excluding Akqi) in Xinjiang, China. The area is approximately $3.6 \times 10^4$ km$^2$, including six counties or cities in the Aksu area (Aksu, Wensu, Awati, Wushi, Keping and Alar). It is served by the western upper reaches of the Tarim River Basin (Fig. 1) and sand dunes are the predominant landform (*El-Tantawi et al., 2019*). The water used for agricultural irrigation makes up more than 95% of the total regional water consumption. The most important irrigated crop is cotton, which has increased in annual planting area (*Li et al., 2020a*; *Li et al., 2020b*).

### Data sources

Meteorological data were obtained from the China Meteorological Science Data Network (1961–2005) (http://data.cma.cn/). Large-scale climate variables (predictors) for the current climate and future scenarios under the RCPs in years 1961 to 2050 obtained from Canadian Climate Data and Scenarios (http://climate-scenarios.canada.ca/). We used grid resolution 2.815° latitude by 2.815° longitude. River discharge data (1961–2005) were from the Aksu
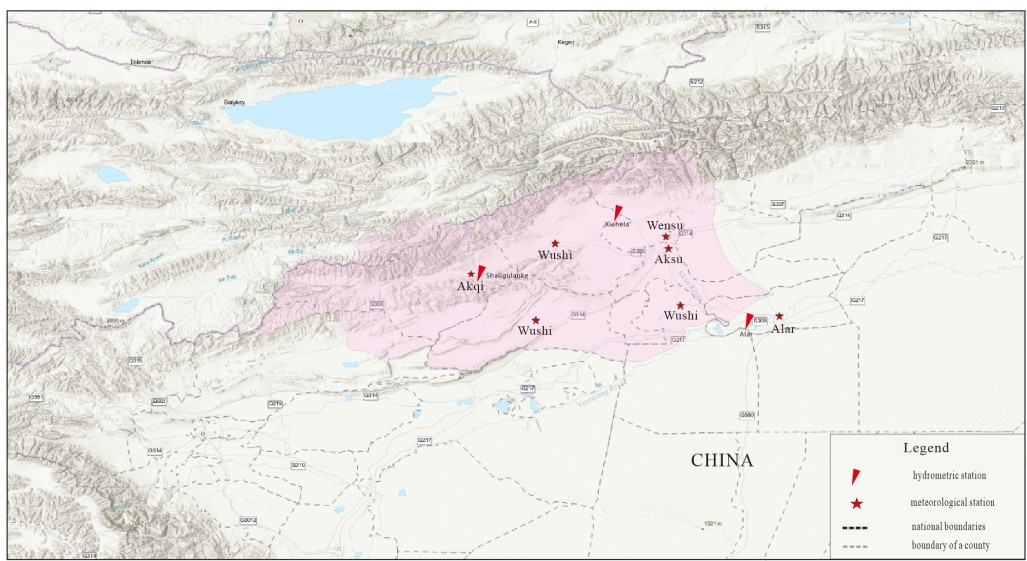

Figure 1 **Location of study area.**

Valley Chronicles (*Aksu River Basin Management Office, 2006*). Socio-economic data were gathered from the Aksu Region Yearbook and Xinjiang Construction Corps Yearbook (*Bureau of Statistics of Xinjiang Production and Construction Corps, 2009*).

## Simulation of climate scenarios
### Statistical downscaling model under two scenarios (RCP 4.5 and RCP 8.5)

RCP 2.6 represents a stringent mitigation scenario, RCPs 4.5 is intermediate mitigation scenarios and RCP 8.5 is low mitigation scenario with very high greenhouse emissions (*Carvalho et al., 2019*). Due to better representation of actual emissions since 2000 by other RCPs (*Peters et al., 2012*), we excluded RCP 2.6 in this study. The CanESM2 predicators provide 26 parameters (https://climate-scenarios.canada.ca/?page=pred-canesm2). To produce climate data for future analyses, the SDSM model was parameterized by inputting daily observations (Fig. 1) and 26 predictors from CanESM2 (1961–1990 data for model building and 1991–2005 data for the model validation). Five predictors (daily maximum and minimum temperature, daily relative humidity, annual rainfall, and annual sunlight) were selected based on the correlation matrix, partial correlation, and *P*-value (*Fowler, Blenkinsop & Tebaldi, 2007*) (Table S1). Final model accuracy was examined using both the coefficient of determination ($R^2$) and Root Mean Square Error (RMSE) (*Wood et al., 2004*) (Table S1).

## Estimation of water demand and supply in Aksu River Basin
### Total water demand

*Agricultural water demand.* The Hargreaves equation and downscaling simulation results were used to calculate the reference crop evapotranspiration ($ET_0$). Previous research showed that Hargreaves equation had good applicability in arid and semi-arid regions

(*Wang et al., 2013a*; *Wang et al., 2013b*).

$$ET_0 = \frac{K}{\lambda}(T_{max} + T_{min})^n \cdot (T_{mean} + T_{off}) \cdot R_a \tag{1}$$

$$T_{mean} = (T_{max} + T_{min})/2 \tag{2}$$

where $K$ is the conversion coefficient (recommended value $= 0.0023$), $\lambda$ is the latent heat of water vaporization (recommended value $= 2.45$ MJ/kg), $T_{max}$, $T_{min}$ are the highest and lowest temperature ($°C$), $n$ is the exponential coefficient (recommended value $= 0.5$), $T_{mean}$ is the average temperature ($°C$), $T_{off}$ is the temperature constant (recommended value $= 17.8$), and $R_a$ is the solar insolation at the top of the atmosphere MJ/(m$^2$/d) (*Bautista, Bautista & Delgadocarranza, 2009*). The water requirement of the main crops in the study area was calculated as:

$$W_{GD} = \sum_{i=1}^{n}\sum_{j=1}^{m} P_i(ET_{0ij}K_{cij} - 0.52T_{ij}) \tag{3}$$

where $W_{GD}$ is the water requirement per unit area of arable land ($10^4 m^3/km^2$), $P_i$ is the proportion of crop i per unit area of arable land, $ET_{0ij}$ is the reference crop water requirement for crop i in month j (growing season), $K_{cij}$ is month j (growing season) of crop i, $T_{ij}$ is month j (growing season) of crop i rainfall, and 0.52 is the rainfall utilization coefficient (*Rahman, Islam & Hasanuzzaman, 2008*).

Irrigation water requirement per unit area:

$$W_{GDi} = W_{GD} * a_i/b_{i1} + W_{GD} * (1 - a_i)/b_{i2} \tag{4}$$

where $W_{GDi}$ is the amount of irrigation water per unit of arable land, $a_i$ isthe proportion of water-saving irrigation area in year i, and $b_{i1}$ and $b_{i2}$ are respectively the conventional and water-saving irrigation water utilization coefficients in year i.

*Industrial and domestic water demand ($W_{IDi}$).* According to 2018 statistics data, industrial water consumption per km$^2$ was calculated by dividing the total industrial outputs with total industrial water consumption in Aksu. The water consumption of the industrial added value of ten-thousand yuan was approximately 110 m$^3$. Residential water consumption per unit area is calculated using the resident population and the water consumption per municipality. According to the Plan for Reform and Development of the Aksu Region (2020–2050), and these are summed for the Aksu River Basin (Table 1).

*Ecological water demand.* We consider water for the forests and water bodies as ecological. The formula for the calculation of forest demand is:

$$W_{LD} = K_s \sum_{j=1}^{m}(ET_{0j}K_{cj} - 0.52T_j) \tag{5}$$

where $W_{LD}$ is the water demand per unit area of woodland in the watershed ($10^4 m^3/km^2$), $K_s$ is the soil moisture limitation coefficient, $ET_{0j}$ is month j of the forest land (growing

**Table 1  Water demand for construction land per unit area (104 m³/km²).**

| Administrative regions | Wushi | Wensu | Keping | Awati | Aksu | Alar |
|---|---|---|---|---|---|---|
| 2020 | 11.88 | 23.20 | 16.48 | 6.14 | 17.40 | 10.45 |
| 2035 | 16.45 | 30.84 | 25.13 | 9.45 | 29.20 | 23.57 |
| 2050 | 22.11 | 25.31 | 21.19 | 8.50 | 23.81 | 18.18 |

season) reference crop water demand, $K_{cj}$ is the crop coefficient of forest land in month j, and $T_j$ is the rainfall of forest land in month j.

$$W_{LDi} = W_{LD}/b_{i2}. \tag{6}$$

Here, $W_{LDi}$ is the amount of irrigation water per unit of woodland, and $b_{i2}$ is the water-saving irrigation water utilization coefficient in decade i.

The water demand for the water bodies:

$$W_{WA} = 0.58 * E_0 \tag{7}$$
$$ET_0 = 0.556 * E_0 \tag{8}$$

where $W_{WA}$ is water demand per unit of water area. *Xi & Cheng (2002)* estimated the conversion coefficient between a 20 cm² dish and a 20 m² evaporating pool was 0.58. $ET_0$ was estimated by multiplying $E_0$ by a coefficient 0.556 (*Xi & Cheng, 2002*). We assume that grasslands are not irrigated in this study.

Total water demand = total agricultural water demand + total industrial and domestic water demand + total ecological water demand.

### Total water supply in the future climate

Total water supply = available surface water + available groundwater resources. The sum runoff data of two hydrological stations (Sahliguilanke and Xiehela) was used as water resource input and the river flow at the Alar station was used as the residual amount of water resource (Fig. 1). The difference between the two flows was computed as the amount of available surface water in the study basin. A neural networks model was used to estimate runoff data for the hydrometric station.

The specific analysis we used followed *Zarghami et al. (2011)*. Using the runoff and meteorological data from 1958–1995, the feedforward neural network models between runoff and meteorological factors were parameterized. Data from 1996–2003 were selected for verification and the fluctuation and precision judgment indices were set to evaluate neural network performance. Because the number of network layers was 20, the two indexes reach the minimum value by trial-and-error. The following data were used in the models: runoff (R), precipitation (P), relative water content (RWC), minimum temperature ($T_{min}$), maximum temperature ($T_{max}$), average daily sunshine hours (ADS), mean temperature ($T_{mean}$) and $ET_0$, all indexed to time (years). The resulting model is:

$$R(t) = f(P(t), RWC(t), T_{max}(t), T_{min}(t), T_{mean}(t)ADS(t)ET_0(t)). \tag{9}$$

The modeling process was performed using the ANN toolbox in the MATLAB environment. Annual average meteorological values predicted by SDSM under RCP4.5 and RCP8.5 scenarios were entered into the neural network model to estimate the annual runoff at the hydrological station. Finally, an expected future amount of water resources could be calculated from the estimated hydrological station data. The available amount of groundwater in Aksu River Basin was considered as unchanged.

## Multi-objective optimal allocation model of water and soil resources

The multi-objective optimal allocation model must cover the balance of the economic, social and ecological benefits. We used the gross national product (GDP) as the economic indicator, the maximum benefit of water per cubic meter as the social indicator and ecological green equivalent as the ecological indicator. The computational formula is:

$$F_1(X) = \max \sum_{i=1}^{n} \sum_{j=1}^{m} a_{ij} X_{ij} \tag{10}$$

$$F_2(X) = \max \frac{\sum_{i=1}^{n} \sum_{j=1}^{m} a_{ij} X_{ij}}{\sum_{i=1}^{n} \sum_{j=1}^{m} b_{ij} X_{ij}} \tag{11}$$

$$F_3(X) = \max \sum_{i=1}^{n} \sum_{j=1}^{m} c_{ij} X_{ij} \tag{12}$$

$$\sum_{i=1}^{n} b_{ij} X_{ij} \leq W_s \tag{13}$$

$$\sum_{i=1}^{n} X_{ij} = T \tag{14}$$

$$\sum_{j=1}^{n} X_{1j} \geq PL_{min} \tag{15}$$

$$\sum_{j=1}^{n} X_{2j} \geq FL_{now} \tag{16}$$

$$\sum_{j=1}^{n} X_{3j} \geq CL_{now} \tag{17}$$

$$\sum_{j=1}^{n} X_{4j} \geq WL_{now} \tag{18}$$

$$\sum_{j=1}^{n} X_{5j} = NL_{now} \tag{19}$$

where $F_1(X)$ is total GDP, $F_2(X)$ is the utilization of maximum benefits per cubic meter of water, $F_3(X)$ is the ecological green equivalent of the river basin, $X_{ij}$ is the area of land types in each area (km$^2$), $a_i$ is gross national product per unit area of each land type (10,000 yuan/km$^2$), $b_i$ isthe water demand per unit area of each land use type (m$^3$), and $c_i$ is the green equivalent value of each area of each land type. T is the total area (km$^2$), $PL_{min}$ is the

**Table 2  Pairwise comparison scale for analytic hierarchy process (AHP) preferences.**

| Definition | Equally important | Moderately important | Strongly important | Verystrong important | Extremely important |
|---|---|---|---|---|---|
| Numerical | 1 | 3 | 5 | 7 | 9 |

**Table 3  AHP calculation results.**

| | Weight of each target | | | Test rating | |
|---|---|---|---|---|---|
| | Economic benefit | Social benefit | Ecological benefit | CI | CR |
| 2020 | 0.6267 | 0.0936 | 0.2797 | 0.0429 | 0.0825 |
| 2035 | 0.5695 | 0.0974 | 0.3331 | 0.0123 | 0.0236 |
| 2050 | 0.5273 | 0.0992 | 0.3735 | 0.0018 | 0.0036 |

red line of cultivated land in the study area (km$^2$), $FL_{now}$ is the current forest area (km$^2$), $BL_{now}$ is the current construction land area, $WL_{now}$ is the current water area, and $NL_{now}$ is the current unused land area.

## The MOGWO algorithm and the optimal solution
### Design of MOGWO algorithm
We modified the grey wolf algorithm (GWO) to incorporate two new components (storing non-dominated Pareto optimal solutions archive and a leader selection strategy) to comprise the MOGWO (*Mirjalili et al., 2016*). The detailed procedures are described in Fig. 2.

We obtained a set of non-dominated solutions for the multi-objective model. The parameter settings were number of wolves = 100, achieve = 100, range = 20% and iterations = 100.

### The best optimal value for multi-objective model based on analytic hierarchy process method (AHP)
To establish a judgment matrix, the relative weight of each target is determined. At the same time, the consistency index of the judgment matrix is calculated to verify the validity of the weight. The consistency of the matrix is considered acceptable when the consistency ratio (CR) is less than 0.1. We determined importance indicators for establishing the judgment matrix (Table 2) and the proportions and test indicators of the three planning goals calculated by the analytic hierarchy process (Table 3).

## RESULTS
### Projected future climate and water resource change
We projected temperatures and precipitation at the Aksu Basin using the downscaled global climate models (GCMs) (Fig. 3). Warming was predicted for this area's subregions. By the year 2050 (starting in 2020), the projected temperature could increase up to 1.3, 0.9, 0.8, and 0.2 °C, at Aksu, Keping, Alar, and Akqi under RCP4.5, respectively. Under the RCP8.5 climate scenarios, the temperatures were predicted to increase at a faster rate. As opposed

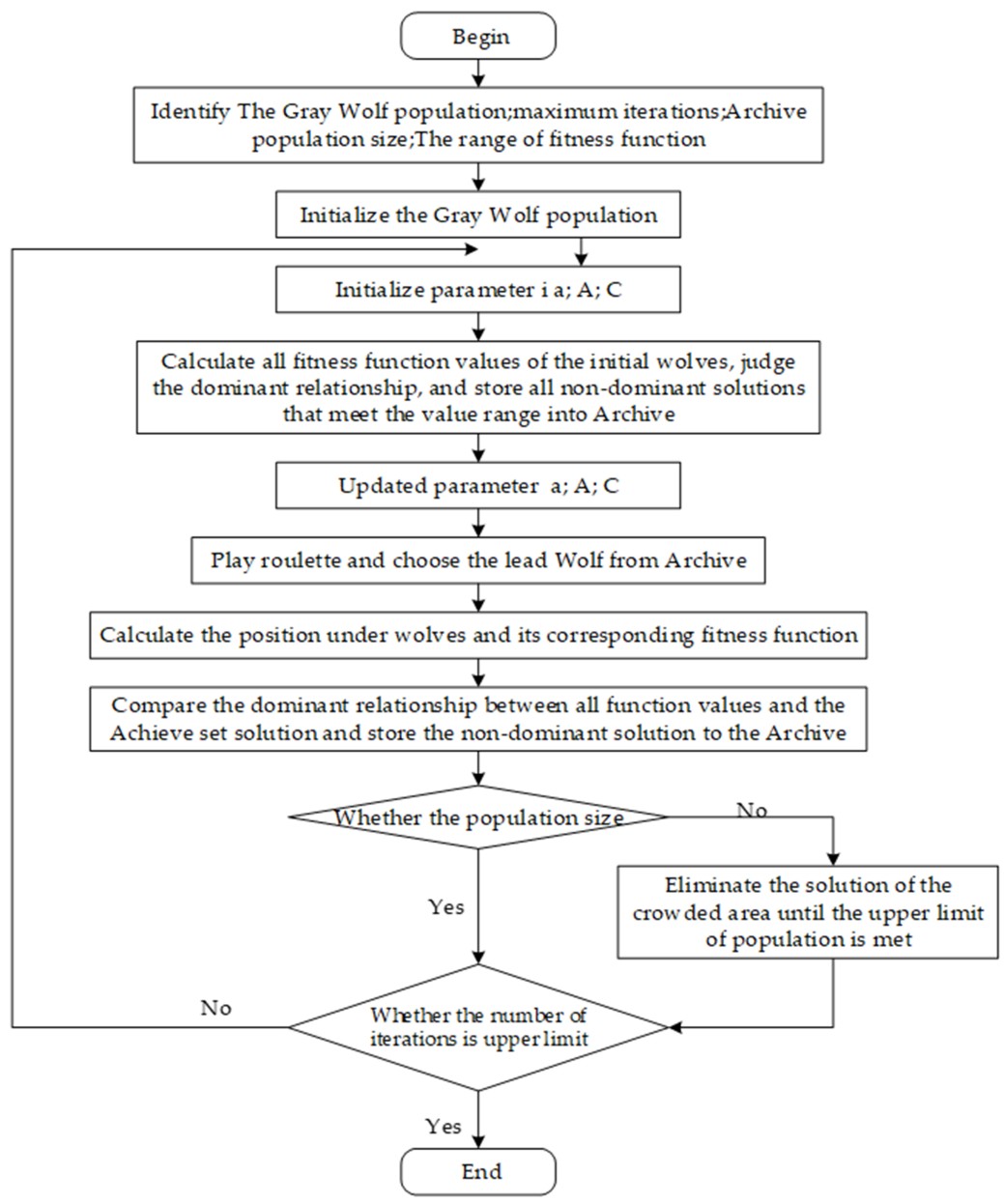

**Figure 2 Flowchart for the MOGWO algorithm.**

to temperature trends, the precipitation showed decreasing trends except at the Keping station.

The minimum and maximum temperatures predicted by SDSM from 2006 to 2050 change over the whole basin's $ET_0$ (Figs. 4 and 5). $ET_0$ values of the four weather stations showed an upward trend during the period 2010–2050. There was no significant difference between the RCP4.5 and RCP8.5 climate scenarios during the first years, but $ET_0$ has

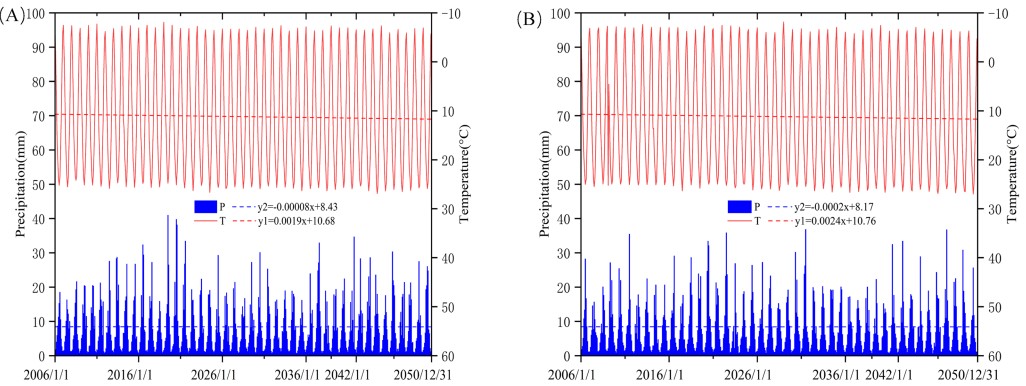

**Figure 3** Average changes of temperature (T) and rainfall (R) of the basin under RCP4.5 (A) and RCP8.5 (B) scenarios during 2006–2050.

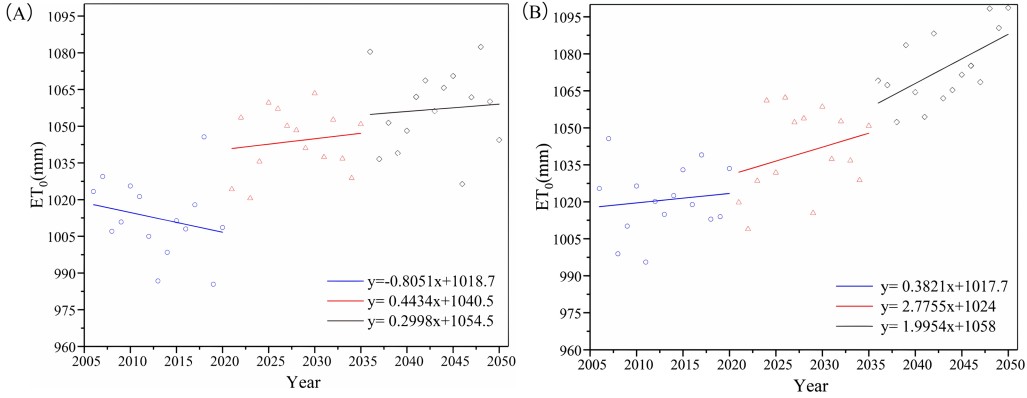

**Figure 4** Average changes of reference evapotranspiration (ET0) of the basin under RCP4.5 (A) and RCP8.5 (B) scenarios during 2006–2050.

an increasingly higher value under RCP8.5 scenarios relative to RCP4.5 after 2035. The difference value is expected to be as high as 50 mm by 2050.

By using the neural network model to estimate the runoff flow of the hydrological station, the available surface water in the basin gives a trend of slow future increase (Fig. 6). Until 2050, the annual average run off is predicted to increase $7.963 \times 10^8$ $m^3$ and $10.41 \times 10^8$ $m^3$ under RCP4.5 and RCP8.5, respectively. The amount of runoff was higher under the RCP8.5 scenario than the RCP4.5 scenario (Table 4).

## The optimal allocation of the water and land resources

Future water shortage is predicted to be about $5.83 \times 10^8 m^3$ in the basin (Table 5). The Pareto frontier under the five scenarios obtained by MOGWO is shown in Fig. S1; the specific values of water and soil resource allocation in the basin are shown in Tables S2–S7. Due to the high emission concentration (RCP8.5), with the exception of construction land, the water demand per unit area of the other land types is higher compared to the low

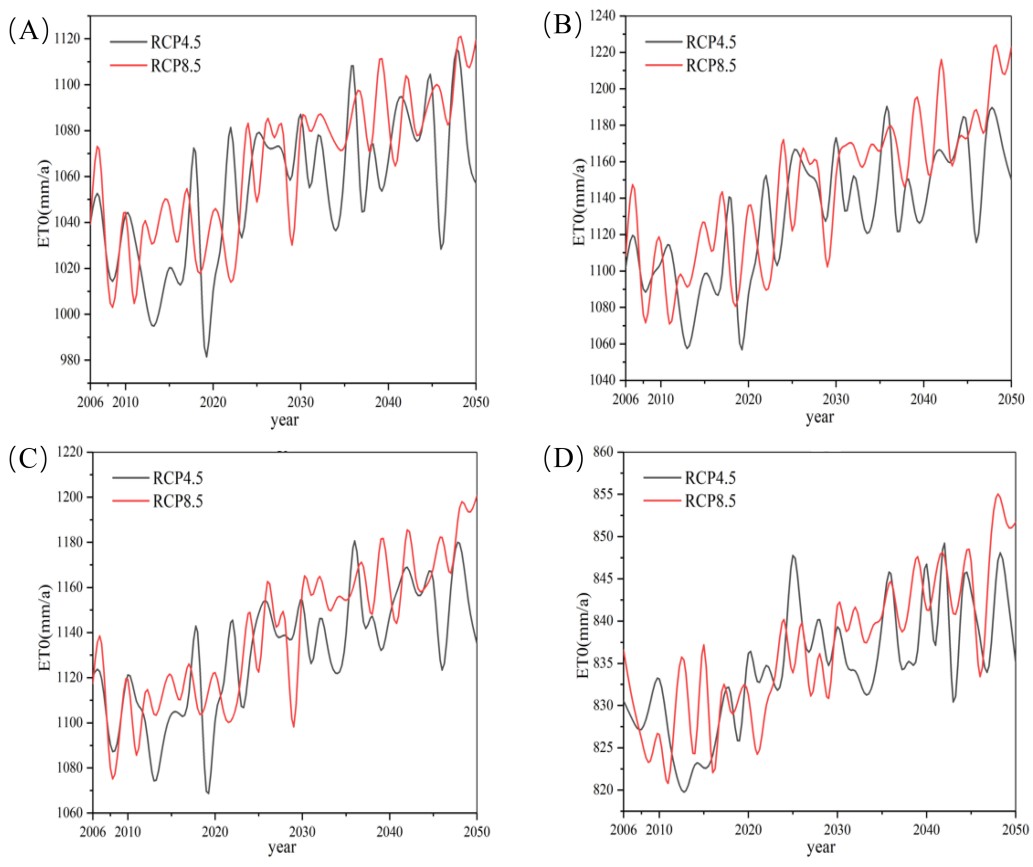

**Figure 5** Average changes of reference evapotranspiration ($ET_0$) of the basin under RCP4.5 and RCP8.5 scenarios during 2006–2050: (A–D) Aksu, Keping, Alar, and Akqi, respectively.

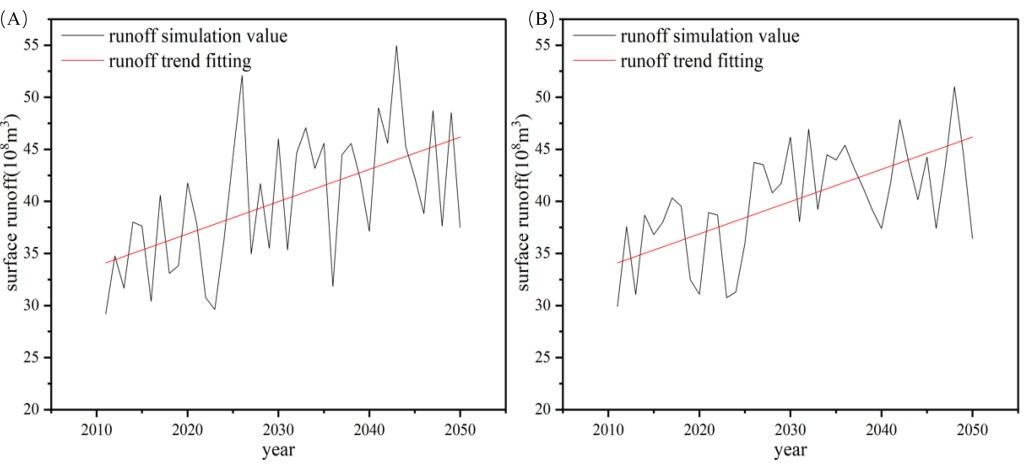

**Figure 6** Neural network predictions of runoff: (A) RCP4.5; (B) RCP8.5.

**Table 4  Prediction results of basin water resource availability.**

| Climate scenarios | | | RCP4.5 | | RCP8.5 | |
| --- | --- | --- | --- | --- | --- | --- |
| Year | 2020 | 2035 | 2050 | 2035 | 2050 | |
| Available amount of water resources($10^8 m^3$) | 47.20 | 50.48 | 51.17 | 52.36 | 53.61 | |

**Table 5  Optimal solution objective function value and water resource supply and demand in RCP4.5 and RCP8.5.**

| | | Economic benefit GDP ($10^8$ yuan) | Social benefit, Unilateral water benefit (yuan/m³) | Ecological benefit, green equivalent (km²) | Water demand $10^8 m^3$ | Available water $10^8 m^3$ |
| --- | --- | --- | --- | --- | --- | --- |
| 2018 | Actual | 544.21 | 10.47 | 8991.40 | 51.87 | 46.04 |
| 2020 | Recent | 613.53 | 13.00 | 9011.21 | 47.19 | 47.20 |
| 2035 | RCP4.5 | 1833.82 | 36.33 | 8999.27 | 50.47 | 50.48 |
| | RCP8.5 | 1836.32 | 35.89 | 9004.21 | 51.17 | 51.17 |
| 2050 | RCP4.5 | 4909.23 | 97.47 | 9164.84 | 50.37 | 52.36 |
| | RCP8.5 | 4910.22 | 96.93 | 9168.69 | 50.66 | 53.61 |

emission concentration (RCP4.5). Although the total cultivated area in the basin under the two climate scenarios is similar, the change of cultivation in each region of the basin differs. Water and land resource allocation in the recent-term (2020), medium-term (2035), and long-term (2050) plan under two emission concentrations showed, in general, a trend of decreasing arable land and grassland and increasing other land (Fig. 7). The arable land areas of Awati, Aksu, and Alar exhibited a continuous downward trend as the result of the policy for restoring farmland to save water, but the arable land areas was likely to continue to increase in Wushi County (Fig. 8). In addition, the area of grassland in the Wushi and Wensu regions was trending downward, while the Alar was increasing (Fig. 9).

## DISCUSSION

We found that SDSM had higher predictive accuracy for temperature relative to rainfall using a formal accuracy index. This may mimic the model's limitations in simulating rainfall (*Wilby, Dawson & Barrow, 2002*). Downscaled climate change model scenarios suggestted that the warm-wet climate trend would continue in the semi-arid region. Rainfall showed a declining trend (except for Keping station) in the region during 2021–2050, which had been also found in other studies (*Chu et al., 2010*; *Wilby & Dawson, 2013*; *Zhu et al., 2019*).

The change of $ET_0$ caused by climate change would have a significant impact on agricultural and ecological water demand (*Guo & Shen, 2016*; *Hadinia, Pirmoradian & Ashrafzadeh, 2016*). $ET_0$ values in the Aksu River basin trend upward in the future (Fig. 6) as in other areas of China such as the Tibetan Plateau, Haihe River Basin and Hetao Irrigation District (*Wang et al., 2013a*; *Wang et al., 2013b*; *Xing et al., 2014*; *Zhou et al., 2017*). The increasing $ET_0$ rate was also inversely related to elevation (*Zou et al., 2020*), therefore $ET_0$ of the Akqi station had the lowest increasing rate in Aksu River basin.

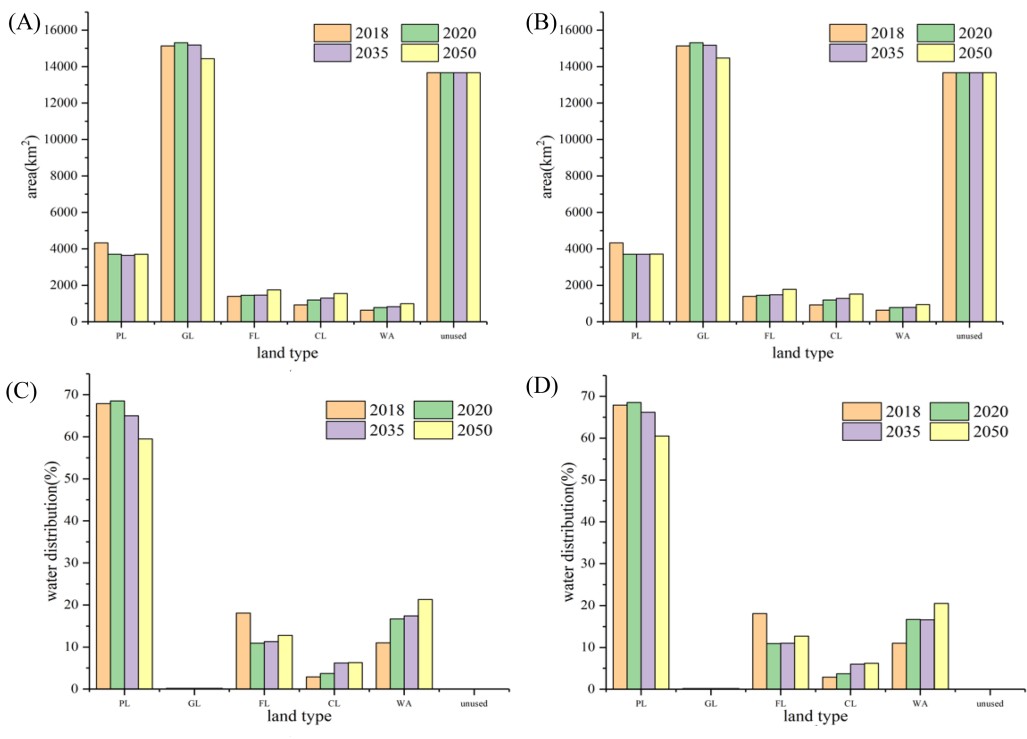

**Figure 7** Allocation of water and soil resources throughout the basin: (A) RCP4.5 land resources; (B) RCP8.5 land resources; (C) RCP4.5 water resources; (D) RCP8.5 water resources. PL, plowland; GL, grassland; FL, forest land; WA, water area.

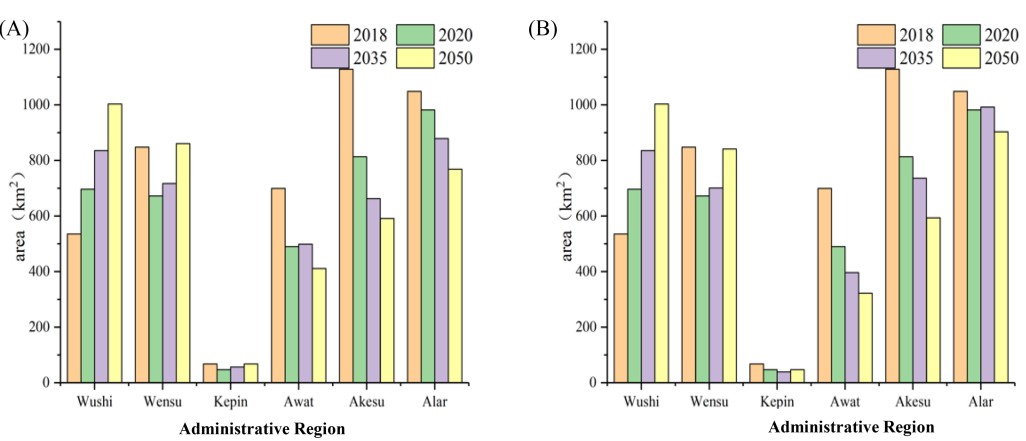

**Figure 8** Allocation of plowland in each county in the basin under RCP4.5 (A) and RCP8.5 (B).

Because of the hydrology in Xinjiang under future climate change scenarios (*Li et al., 2020a*; *Li et al., 2020b*; *Shen et al., 2020*; *Xu et al., 2010*), we suggestted that surface runoff in the basin would trend upward in the future. We suspected that the main reason for increased runoff was an increase in temperature leading to amplified loss of seasonal snow

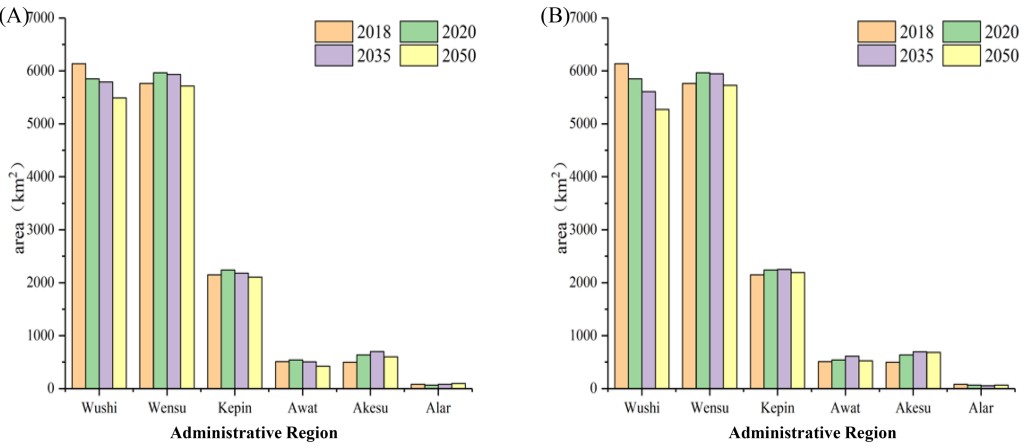

**Figure 9  Allocation of grassland in each county in the basin under RCP4.5 (A) and RCP8.5 (B).**

cover, glacier and ice sheet melting (*IPCC, 2021*). In the past 30 years, ice and snow melt water had been increasing and would continue to increase until 2050 in the Aksu River Basin (*Wang, 2018*; *Zhang, 2010*), which also supportted our model results. *Ding & Reng (2007)* reported that glacier melt water around the Tarim Basin would continually grow potentially reaching a level of $10^8$ m$^3$/a (*Xia Jun, 2011*). The upward trend in runoff was also predicted to be greater under RCP 8.5 than RCP4.5 (Fig. 6).

Water scarcity continues to be a major crisis in the Aksu River Basin. With the $ET_0$ increased, water demand from agricultural production and ecological protection had increased annually. The multi-objective allocation of soil and water resources to economic, societal, and ecological goals, compelled us to recommend reducing the area of cultivated land to alleviate the current water shortage (*Xu et al., 2010*). Currently, the Aksu River Basin is facing a shortage of water resources estimated to be $5.83 \times 10^8$ m$^3$ (Table 5). To balance water supply and demand, the Aksu River Basin needs to reduce agricultural water use as a measure to protect the environment. The most effective mitigation action is to reduce the arable land area. Forests and water bodies should be expanded to strengthen ecological protection and improve ecosystem services. Therefore, regions should adopt alternative allocation strategies to achieve the optimal comprehensive benefits for the whole basin. Decision makers should enact recommended configurations according to their own conditions under a changing climate and in different regions.

Notably, the change of cultivation in each region of the basin differs. The arable land areas of Awati, Aksu, and Alar exhibited a continuous downward trend as the result of the policy for restoring farmland to save water, but the arable land areas were likely to continue to increase in Wushi County (Fig. 8). The reason for this heterogeneity may be the different water requirements per unit area of arable land in each county under the two scenarios. The output value per unit area of Wushi County was lower than other regions due to the large proportion of crops planted on arable land. As well, water demand per unit area was smaller than in other regions. If water shortages constrained watershed development and are red line constraint of arable land, then lower water use in Wushi would have a

greater impact on the overall benefits of the entire watershed than a low output value. We provided the following policy recommendations: To cope with this shortage in the basin and measure including strictly following the "red line" restriction of cultivated land, the area of the cropland should be reduced by 127.5 km$^2$ under RCP4.5 or 377.2 km$^2$ under RCP8.5 models. For the sake of ecological sustainability, the allocation ratio of forest and water bodies should increase to 39% of the total water volume in the Aksu River Basin by 2050.

There is some limitation in this study, and the model could still be improved. For example, due to the lack of data on future the cultivated land planting structure, we used current cultivated planting structure for calculating future water demand per unit area. And in the AHP method, the interconnection between the factors in the criterion level was ignored and considered as independent of each other, and the factors could be refined in the later study to make the results closer to the actual situation.

## CONCLUSIONS

In the future, the $ET_0$ of the Aksu River Basin would increase variably according to the climate predictions of an SDSM model. Our study indicated that the water resources, mainly generated by glacier/snow meltwater, increased according to the neural network model. Climate change may have beneficial effects on agriculture in Aksu River Basin. This outcome may force governments to find new and sustainable adaptation strategies to rescue the future water supply. The water governance in this region should be more flexible and decentralized to cope with climate change.

## ACKNOWLEDGEMENTS

We are very thankful to Dr. John Wilmshurst for carefully reading and polishing the manuscript.

### Funding

The research was supported by the Grant from the National Key Research and Development Program of China (2016YFC0400208), the National Natural Science Foundation of China (Grant No. 51809269), the Farmland Irrigation Research Institute of Chinese Academy of Agriculture Co-ordination Project (FIRI2022-02) and the National Cotton Industrial Technology System (CARS-15). The funders had no role in study design, data collection and analysis, decision to publish, or preparation of the manuscript.

### Grant Disclosures

The following grant information was disclosed by the authors:
National Key Research and Development Program of China: 2016YFC0400208.
National Natural Science Foundation of China: 51809269.

Farmland Irrigation Research Institute of Chinese Academy of Agriculture Co-ordination Project: FIRI2022-02.
National Cotton Industrial Technology System: CARS-15.

## Competing Interests
The authors declare there are no competing interests.

## Author Contributions
- Zhidong Wang performed the experiments, analyzed the data, prepared figures and/or tables, authored or reviewed drafts of the article, and approved the final draft.
- Xining Zhao conceived and designed the experiments, analyzed the data, authored or reviewed drafts of the article, and approved the final draft.
- Jinglei Wang conceived and designed the experiments, authored or reviewed drafts of the article, and approved the final draft.
- Ni Song performed the experiments, prepared figures and/or tables, and approved the final draft.
- Qisheng Han conceived and designed the experiments, performed the experiments, analyzed the data, prepared figures and/or tables, authored or reviewed drafts of the article, and approved the final draft.

## Data Availability
The raw measurements are available in the Supplementary Files.

## Supplemental Information
Supplemental information for this article can be found online at http://dx.doi.org/10.7717/peerj.14577#supplemental-information.

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
