# Peer review of "Agricultural water allocation with climate change based on gray wolf optimization in a semi-arid region of China"

_PeerJ, doi:10.7717/peerj.14577_

## Round 0.1 · original submission · Major Revisions

Please revise according to the comments of reviewers.

Our apologies for the slower than expected decision - two reviewers did not provide their promised reviews.

Reviewer 1 ·

Basic reporting

About the use of English, the article is understandable for non-English native speakers. However, in the following lines 59, 68, 69, there are some words with written errors.

About Table -1
Line 162: “Table 1 is the sum of industrial and domestic water consumption for the Aksu River Basin.”
In Table 1, it’s not possible to understand the unit expression of that consumption. Previously, it said (Lines 160 -161): “Industrial water consumption per unit area is calculated by multiplying water consumption per 10,000 yuan of output value by the industrial output value”.

About literature references: The references that were used are relevant to the article.

The article gives answers to the hypotheses and proffers a relevant analysis to decision makers about the consequences under the two RCP scenarios considered.

Experimental design

Line 125, this link is broken (https://climate-scenarios.canada. 126 ca/?page=pred-canesm2)

When in the article explain SDSM model, in line 129, the article says that five predictands were used, but in the Table 1 - Supplementary, only appear three predictands. Which is the reason?

In the article the other materials and methods are rigorously explained.

Validity of the findings

The integration of the models used is novel and can be taken as a reference for studies aimed at territorial planning under climate change scenarios.

Finally, I have a question. If the largest amount of water in the basin comes from the melting ice of the glaciers and according to evidence these tend to reduce (329-332, 349-351, and another research 370-371), same as rainfall (line 313), Why was the reduction in snow cover not considered for the allocation model?

Additional comments

Relevant study to territorial planning and for climate change adaptation

Annotated reviews are not available for download in order to protect the identity of reviewers who chose to remain anonymous.

·

Basic reporting

This paper is well structured and background is sufficient.

Experimental design

The method is adequately described but sitll need some minor revision.

Validity of the findings

Discussion and conclusion can be improved, please see details in my additional comments.

Additional comments

The authors aim to resolve the allocation problem of agricultural water under future climate change scenarios. They deploy GWO to solve the multi-objective programming. The topic is classic and the authors properly integrate climate change with optimal allocation of agricultural water. The research design is sufficient. However, before it is considered publishable, some issues should be addressed.

1. Some sentences in abstract like line 33-34; line 39-40 should be rephrased.
2. There are some minor language issues/typo/confusing sentences, like line 85, China; line 309-310; Please make careful proofreading throughout the manuscript.
3. The optimization problem under certain resources constraint has been researched to death in many disciplines. So, in this paper, you should further highlight the new thing you contribute to existing knowledge. Now, in introduction part, this is not adequately described.
4. For some coefficients of the formulas (e.g. line 142-147) in the method part, do you subjectively determine them? If no, any reference? what is the applicability in your case? Also, please number each formula.
5. Line 208, as to the social indictor, it indeed reflects the water use efficiency, so how can it be linked to social aspect? Line 225, the unit yuan/m^3 should be m^3. Please carefully collate each formula and the variable description because there are stacks of formulas.
6. Do you compare your results with the traditional GWO to observe the improvement of your new MOGWO method? You may do this.
7. You only design two future climate scenarios RCP4.5 and RCP8.5, I suggest you incorporate RCP2.6, which is a least climate stress scenario. This may facilitate insightful comparison. Also, I see the predicting power for rainfall is relatively low (R-square is about 0.3), why and how will this influence the uncertainty for analysis?
8. Line 238-243, AHP is a weight determining method relying on experience, which is also a subjective method? Why you say it can avoid the influence of DM’s subjective choice.
9. Line 263, please explain why the increase trend will occur?
10. Line 289-292, please make these sentences clearer.
11. Some explanation (reasons of finding) in the results part should be moved to discussion.
12. Line 344-346, reducing crop land may threaten the food security, please consider this and provide some comments when you propose such policy implication.
13. Limitation should be briefly referred.

---

## Round 0.2 · Minor Revisions

Thank you very much for the large number of modifications made by the author. It is suggested that the article be modified and accepted.

·

Basic reporting

no comments

Experimental design

no comments

Validity of the findings

no comments

Additional comments

Thanks for your revision work. The paper now improves a lot.

Please confirm that you have proofread you text. The authors should still make careful language correction, e.g., line 92-94, line 328
Line 103-109, these statements regarding your so-called contribution is more like the research aims. You should comment what things does your work contribute to existing knowledge in terms of theory or practice.

As to the AHP, you should not just say “we have removed this sentence in the revised manuscript.” You should mention something in the text about this issue.

---

## Round 0.3 · accepted · Accept

Thank you very much for the author's careful revision of each opinion, I suggest accepting and publishing.